# Young-parent communication on sexual and reproductive health issues among young female night students in Amhara region, Ethiopia: Community-based cross-sectional study

**Gedefaye Nibret Mihretie** *, **Tewachew Muche Liyeh** , **Yitayal Ayalew Goshu**, **Habtamu Gebrehana Belay**, **Habitamu Abe Tasew**, **Abeba Belay Ayalew**

Department of Midwifery, College of Health Sciences, Debre Tabor University, Debre Tabor, Ethiopia

* gedefayen@gmail.com

**Data Availability Statement:** All relevant data are within the paper.

## Abstract

### Background

Young is a key stage in rapid biological and psychosocial changes affecting every aspect of the lives and an important time to set the foundation for good health in adulthood. Adolescent-parent communication is a potential path for improving sexual and reproductive health outcomes for adolescents, most of parents did not teach their adolescents about sexual and reproductive health. Even though, some researches have been done on day time students, there is no study conducted focusing on young girls attending night school in Ethiopia.

### Objective

This study aimed to assess young-parent communication on sexual and reproductive health issues and associated factors among night female students in Amhara Region, Ethiopia, 2018.

### Method

School based quantitative cross-sectional study was employed in Amhara region among 1640 young female night students from September 15 to November 15/2018. Face-to-face interview-administered questionnaires were used to collect the data. Bi-variable and multi-variable logistic regression model were used. Odds ratio (OR) with 95% confidence interval (CI) were computed to determine the strength of association between predictor and outcome variables. P-values less than 0.05 considered as level of significance.

### Results

One hundred ten (37.5%) of the students had communication with their parents on at least two issues of sexual and reproductive health (SRH) issues in the last 6 months. Grade level (adjusted odd ratio (AOR) = 2.61, 95% CI (2.04, 3.34)), marital status (AOR = 1.29, 95% CI

**Funding:** The authors received no specific funding for this work.

**Competing interests:** The authors declare that there is no competing interest.

**Abbreviations:** SRH, sexual and reproductive health; STI, sexual transmitted infections; VCT, voluntary counseling and testing.

(1.03, 1.63), living arrangement (AOR = 1.50(1.13, 2.00)), utilization of youth friendly sexual and reproductive health services (AOR = 1.80, 95% CI (1.41, 2.30)), students ever had sexual intercourse (AOR = 1.50, 95% CI (1.23, 1.96)), Information about sexual and reproductive health services (AOR = 1.45(1.16, 1.80)) were associated young-parents communication on sexual and reproductive health issues.

## Conclusion

In this study young-parent communications on sexual and reproductive health (SRH) issues was found to be limited. Therefore, teachers, health extension workers, and health professionals should strengthen comprehensive SRH education for students in school, churches, mosques, health facilities and encouraging them to participate in different health clubs in school. Parent should give education for their children sexual and reproductive health during the era of young age.

## Introduction

World health organization, young is defined as age group between 10 and 24 years [1]. Young is a key stage in rapid biological and psychosocial changes affecting every aspect of the lives and an important time to set the foundation for good health in adulthood [2]. Early pregnancy and motherhood, difficulty accessing contraception, unsafe abortion, high rates of sexually transmitted infection, including human immunodeficiency virus (HIV), socio-economic and cultural factors are some of the challenges for young people [3]

Globally, one third of women give birth before the age of 20 each year and are at increased risk of morbidity and mortality due to obstetric complications. Annually, 5,000,000 and 70,000 adolescents between the ages of 15 and 18 have unsafe abortions and abortion related deaths, respectively [4]. Sixteen million girls aged 15–19 give birth each year, which is approximately 11% of all births worldwide; 95% of these births occur in low- and medium-income countries [3].

In Ethiopia, adolescents constitute about 24% of the total population and about 60% of adolescent pregnancies are unintended [5,6]. Sexual and reproductive health problems of adolescents are increasing from time to time and this is related with most parents do not feel happy to discuss about sexual issues with their adolescents because they assumed that it leads to early sexual commencement and sexual related diseases [7].

Twenty four percent of women and 2% of men have first sexual intercourse before age 15, 62% and 17% have had sexual intercourse before age 18, respectively. By age 20, 76% of women and 36% of men have had sexual intercourse [8]. Parent-adolescent communication regarding sexual and reproductive health issues is more likely to reduce adolescent risk-taking sexual behaviors. However, Parent-adolescent communication about sexual issues remains a challenging issue in many Sub-Saharan African countries including Ethiopia as the social milieu in many traditional communities still limit such communication [9].

Different cross-sectional studies indicated that sexual and reproductive health of adolescent was 36.67% [10], 28.6% [11] and in Sub- Saharan Africa ranges from 8% to 80% [12]. A studies revealed that mothers educational level [13], socioeconomic status [14], adolescents age [15], grade level [16,17], knowledge [11,18,19] were significantly associated adolescents sexual and reproductive health. Youngs were found information and clues about sexual health from a variety of sources: parents, siblings, peers, magazines, books, and the mass media [20]. Young-

parent communication is a potential path for improving sexual and reproductive health outcomes for adolescents [21]. Parents are primary teachers for their children, and able to direct the progress of children in sexual health matters, enable them to encourage free sexual communication and develop strong personal decision-making skills [11,22].

Even though, some researches have been done on day time students, there is no study conducted concerning young-parent communication on sexual and reproductive health issues focusing on young girls attending night school in Ethiopia. Therefore, assessing young-parent communication on sexual and reproductive health issues and associated factors among female night students are very vital in designing, implementing and monitoring effective health intervention for this particular group.

## Methods

### Study design and setting

School based cross-sectional study was employed from September 15/2018 to November 15/2018 in Amhara region, North West Ethiopia. Amhara region is one of the nine regional states in Ethiopia which is found between 11˚ 30' 00" N latitude & 38˚ 30' 00" E longitude on the Northwestern part of Ethiopia. It is one of Ethiopia's largest regions; it has 11 zones, three city administrations, and 180 districts (139 rural and 41 urban). According to the Ethiopian Central Statistics Agency, the region has a projected population of 21.5 million people, about 80 percent of whom are rural farmers. The region has 80 hospitals (5 referral, 2 general and 73 primaries), 847 health centers, and 3,342 health posts. There are 55 elementary and 23 secondary schools that provide night education.

### Source population

All female night young elementary and high school students in Amhara region were the source population of the study.

### Study population

All young female night students attending elementary and high school in selected zones of Amhara region during the study period considered as the study population.

### Inclusion criteria

Female young students who were attending night school program of elementary and high school and presented during data collection time in Amhara region.

### Exclusion criteria

Female night students who reside for less than six months in selected Zones were excluded.

### Sample size determination

Sample size was determined using single population proportion formula by considering assumptions of proportion of adolescent-parent communication 36.9%, [23] 95% confidence interval, 3% margin of error.

Where $Z_{a/2}$ = 1.96 (with 95% level of certainty).

W = margin of error (3%)

P = 0.369 proportion of adolescent-parent communication.

n = total sample size

$$n = \frac{Z\,a/2\;P(1-P)}{W2} = \frac{(1.96)^2 \text{xo}.369 \text{x} 0.631}{(0.03)^2} = 994$$

Finally, by using design effect 1.5 and 10% of non-response, a total 1640 of participants were included.

## Sampling procedure

Multi-stage sampling techniques were used to select the study participants. First 5 zones were selected by using simple random sampling technique from 11 zones within the region. Then 22 schools were selected by lottery method from each selected zones and sample size was proportionally allocated to all selected elementary and high schools. Finally, study participants were selected by using simple random sampling technique.

## Variables

Dependent variable: Young- parent communication about reproductive health issues. Independent variables: Socio-demographic factors: (age, residence, religion, marital status, Pocket money of students, ethnicity, family size, occupation, grade level of participants, education of father, education of mother, occupation of father, occupation of mother, family income). Reproductive relate factors: (marital-age, unwanted pregnancy, abortion, sexual education, sexual transmitted infections (STI), voluntary counseling and testing (VCT), sexual violence). Individual level and communication related factors: (Knowledgeable on SRH, sources of information).

## Operational definitions

Communication on SRH issues: Students who discussed at least two SRH issues (premarital sex, early marriage, unwanted pregnancy, abortion, sexual transmitted infections, voluntary counseling and testing, family planning) with their parents in the 6 months.

Young: are peoples who are between 10–24-year-olds.

Knowledgeable: Students who score points more than or equal to mean score out of prepared knowledge questions.

Not Knowledgeable: Students who score points less than mean score out of prepared knowledge part questions [17,23].

## Data collection instruments and procedures

The data were collected by interviewer-administered, structured questionnaires adapted from different literature. Thedata collection tools (questionnaires) were first prepared in English and then translated to a local language (Amharic) and then re-translated back to English language by language experts to keep its consistency. The data were collected by 15 BSc midwives trained on how to assist students, take consent and how to monitor the overall data collection process in selected zone of Amhara region. Data collectors were supervised by 10 MSc midwives. Finally, filled questionnaires were signed by supervisors after checking for its completeness.

## Data quality assurance

To assure the quality of the data, technical training was given for data collectors and supervisors for three days. Pre-test was given for 5% of sample size in out of selected schools. During data collection supervision was conducted. After data collection, checking of data entry and

cleaning were conducted for the completeness of the data. Throughout the course of the data collection, interviewers were supervised at each site, regular meetings were held between the data collectors, supervisor and the principal investigator together in which problematic issues arising from interviews were discussed and addressed. The collected data were reviewed and checked for completeness before data entry.

### Data analysis

The data were entered into EPI-Data 3.1 statistical software and then sorted, cleaned, and analyzed by using SPSS version 23 statistical package. Descriptive statistics were done to describe the study population in relation to relevant variables by using text and tables. Bi-variable and multi-variable logistic regression model were used. Odds ratio with 95% confidence interval were computed to determine the strength of association between predictor and outcome variables. P-values less than or equal to 0.05 considered as level of significance.

### Ethical consideration

Ethical clearance for this study was obtained from ethical review committee of Debre Tabor University College of medicine and health sciences and supporting letter was obtained from Amhara Regional Education Bureau. This support letters were sought to each zonal town and forwards to elementary and secondary school on which the studies were conducted. Written informed consent for ≥ 18 years old and assent for <18 years from their families were obtained after explained the purpose and objective of study. Interviews were taken place in a convenient place to maintain privacy and to assured confidentiality.

## Results

### Socio-demographic characteristics of respondents

A total of 1640 school adolescents were involved in the study with 100% response rate. One thousand eighty-eight (66.3%) of the respondents were within the age group of 20–24 years. More than two third of respondents 1071(65.3%) were attending elementary education (1–8 grade). Majority of respondents 1440(87.8%) were Orthodox religion followers and 1519 (92.6%) were Amhara in ethnicity (Table 1).

### Reproductive health factors of the respondents

Nearly half, 802(48.9%) of the students had faced reproductive and sexual health problems at least one of reproductive health problems (unwanted pregnancy, abortion, premarital sex, sexual violence and early marriage). Out of the 1640 respondents, 372(22.7%) had unwanted pregnancy and 695(42.4) had faced sexual violence (Table 2).

### Source of information about sexual and reproductive health

Two third of youths 1061(64.7%) were heard about sexual and reproductive issues. There common source of information were mass media (40.8%), family (24.5%), friend/peers (29.7%) and school (26.2%).

### Adolescent-parent communications on SRH issues

About 37.5% had discussed with their parents about sexual and reproductive health issues. Four hundred ninety-eight of the respondents (30.4%) of the students held discussion with their parents about STI/HIV/ AIDS. Among those students, 1094(66.7%) had not discussed

**Table 1. Socio-demographic characteristic of female night students in Amhara Region, Ethiopia, 2018.**

| Variables(n = 1640) | Frequency(n) | Percentage (%) |
|---|---|---|
| Age | | |
| 15–19 years | 1088 | 66.3 |
| 20–24 years | 552 | 33.7 |
| Grade level | | |
| Elementary school (grade 1–8) | 1164 | 70.8 |
| Secondary school (grade 9–10) | 476 | 29.2 |
| Religion | | |
| Orthodox | 1440 | 87.8 |
| Muslim | 149 | 9.1 |
| Protestant, Catholic | 51 | 3.1 |
| Ethnicity | | |
| Amhara | 1519 | 92.6 |
| Tigre and Oromo | 121 | 7.4 |
| Childhood resident | | |
| Urban | 533 | 32.5 |
| Rural | 1107 | 67.5 |
| Marital status | | |
| Married | 348 | 21.2 |
| Had boy friends | 257 | 15.7 |
| Single | 920 | 56.1 |
| Divorced/widowed | 115 | 7.0 |
| Occupation of participants | | |
| Housewife | 195 | 11.9 |
| Housekeeper/maid | 572 | 34.9 |
| Merchant | 151 | 9.2 |
| Commercial sex workers | 161 | 9.8 |
| Daily laborer | 561 | 34.2 |
| Living arrangement of the respondent | | |
| With family | 276 | 16.8 |
| With partner | 244 | 14.9 |
| With friends | 303 | 18.5 |
| With employer/relative | 390 | 23.8 |
| Alone | 427 | 26.0 |
| Pocket money | | |
| No money | 467 | 28.5 |
| <500 | 326 | 19.9 |
| ≥500 | 846 | 51.6 |
| Maternal education | | |
| No formal education | 932 | 56.8 |
| Primary educational level | 499 | 30.4 |
| Secondary education and above | 210 | 12.8 |
| Maternal occupation | | |
| Housewife | 428 | 26.1 |
| Daily laborer | 745 | 45.4 |
| Merchant | 297 | 18.1 |
| Government employee | 171 | 10.4 |
| Father educational level | | |

(*Continued*)

**Table 1.** (Continued)

| Variables(n = 1640) | Frequency(n) | Percentage (%) |
|---|---|---|
| No formal education | 858 | 52.3 |
| Primary educational level | 556 | 33.9 |
| Secondary education and above | 226 | 13.8 |
| Father occupation | | |
| Farmer | 466 | 28.4 |
| Government employee | 226 | 13.8 |
| daily laborer | 556 | 33.9 |
| Merchant | 392 | 23.9 |

with their parents about unsafe sex (Table 3). Respondents discussed with their mother 371 (79.9%) about early marriage.

## Factors associated with youth-parent communications

Variables with P<0.2 in binary logistic regression model were entered in to multivariable logistic regression. Multivariable logistic regressions revealed that educational status, marital

**Table 2. Reproductive health factors among female night students in Amhara Region, Ethiopia, 2018.**

| Variable (n = 1640) | Frequency | Percentage |
|---|---|---|
| Information about sexual and reproductive health services (n = 1640) | | |
| Yes | 1170 | 71.3 |
| No | 470 | 28.7 |
| Students ever had sexual intercourse (n = 1640) | | |
| Yes | 1170 | 71.3 |
| No | 470 | 28.7 |
| utilization of youth friendly sexual and reproductive health services | | |
| Yes | 1026 | 62.6 |
| No | 614 | 37.4 |
| Age at first marriage (n = 348) | | |
| <18 years | 95 | 27.3 |
| ≥18 years | 253 | 72.7 |
| Faced Unwanted Pregnancy | | |
| Yes | 372 | 22.7 |
| No | 1268 | 77.3 |
| Faced abortion | | |
| Yes | 110 | 6.65 |
| No | 1530 | 93.3 |
| Faced sexual transmitted infection | | |
| Yes | 79 | 4.8 |
| No | 1561 | 95.2 |
| Ever faced Sexual violence | | |
| Yes | 626 | 38.2 |
| No | 1014 | 61.8 |
| Type of sexual violence (n = 626) | | |
| Rape | 117 | 18.7 |
| Unpleasant word | 509 | 81.3 |

**Table 3. Primary and Secondary school night female student's discussion with their parents on SRH issues in the last 6 months in Amhara Region, Ethiopia 2018.**

| Topics of discussion | | With whom they had discussed | | |
|---|---|---|---|---|
| | | **Father or Mother** | **Father*** | **Mother*** |
| STI/HIV | Yes | 498(30.4%) | 286(54.4%) | 312(62.7%) |
| | No | 1142(69.6%) | | |
| Abortion | Yes | 691(42.1%) | 345(49.9%) | 416(60.2%) |
| | No | 949(57.9%) | | |
| VCT | Yes | 729(44.5%) | 534(73.3%) | 415(56.9%) |
| | No | 911(55.5%) | | |
| FP | Yes | 1026(62.6%) | 320(31.2%) | 803(78.3%) |
| | No | 614(37.4%) | | |
| premarital sex | Yes | 546(33.3%) | 456(83.5%) | 432(79.1%) |
| | No | 1094(66.7%) | | |
| Early marriage (age<18) | Yes | 482(29.4%) | 356(73.8%) | 371(79.9%) |
| | No | 1158(70.6%) | | |
| unwanted pregnancy | Yes | 637(38.8%) | 320(50.2%) | 539(84.6%) |
| | No | 1003(61.2%) | | |

*Multiple responses were possible.

status, participants' mother educational status, ever had sexual intercourse, had information about sexual and reproductive health services and living arrangement of participants were significantly associated with youth-parent communication on SRH issues.

Respondents who were attending secondary education were 2.6 times more likely to communicate with their parents on SRH issues as compared with those who were attending elementary education (AOR = 2.61, 95% CI (2.04, 3.34)). Students who had married were 1.29 times more likely to communicate with their parents than those not married (single) (AOR = 1.29, 95% CI (1.03, 1.63). Participants who have lived with employs/relatives were 1.5 times more likely to communicate with their parents than those who have lived alone (AOR = 1.50(1.13, 2.00)).

The odds of having young parent communication were higher for students who utilize youth friendly sexual and reproductive health services (AOR = 1.80, 95% CI (1.41, 2.30)) than the counter parts. Respondents who had ever had sexual intercourse were 1.5 times more likely to communicate with their parents about SRH issues than who had not (AOR = 1.50, 95% CI (1.23, 1.96)). Respondents who had information about sexual and reproductive health services were 1.45 times more likely to communicate with their parents than those who did not heard (AOR = 1.45(1.16, 1.80)) (Table 4).

## Discussion

The aim of this study was assessing young-parent communication on sexual and reproductive health issues and associated factors among female night students. This study showed that 37.5% of students had discussed at least two topics of reproductive health issues with either of their parents in the last 6 months. This finding was lower than the study conducted in Yirgalem (59.1%) [9]. This might be due to participants' grade differences. The study participants of this study were grade 1–10, but the study participants of other studies were grade 9–12. But our study was higher than study conducted in Myanmar (28.6%) [11], Benishangul Gumuz Region (29%) [24], Harar, Ethiopia (28.8%) [22], Awabel (25.3%) [25], Dire Dawa (37%) [18],

**Table 4. Bivariate and multivariable logistic regressions on factors associated with youth-parent y\communications on SRH issues in the last 6 months in Amhara Region, Ethiopia, 2018.**

| Variables | Communication on SRH issues with parents | | COR (95% CI) | AOR (95% CI) |
|---|---|---|---|---|
| | Yes | No | | |
| Age | | | | |
| 15–19 | 347(36.5%) | 266(38.6%) | 1 | 1 |
| 20–24 | 603(63.5%) | 424(61.4%) | 1.09(.89, 1.33) | 1.16(.93, 1.43) |
| Marital status | | | | |
| Single | 181(19.1%) | 111(16.1%) | 1 | 1 |
| Divorced/widowed | 426(44.8%) | 336(48.7%) | .86(.64, 1.15) | .98(.72, 1.33) |
| Married | 343(36.1%) | 243(35.2%) | 1.11(.89, 1.38) | 1.20(1.03, 1.63) * |
| Participant occupation | | | | |
| Housewife | 59(6.2%) | 65(9.4%) | 1.48(1.03, 2.14) | 1.43(.96, 2.13) |
| Housekeeper/maid | 221(23.3%) | 153(22.2%) | .98(.77, 1.26) | 1.00(.77, 1.30) |
| Merchant | 62(6.5%) | 39(5.7%) | .84(.55, 1.27) | .84(.54, 1.31) |
| Employed | 63(6.6%) | 85(12.3%) | 1.90(1.34, 2.69) | 1.50(1.03, 2.19) |
| Commercial sex workers | 18(2.0%) | 12(1.7%) | .85(.413, 1.75) | 1.14(.52, 2.46) |
| Daily laborer | 527(55.5%) | 336(48.7%) | 1 | 1 |
| Grade level | | | | |
| Primary | 612(64.4%) | 552(80.0%) | 1 | 1 |
| Secondary | 338(35.6%) | 138 (20.0%) | 2.20(1.75, 2.77) | 2.18(1.73, 2.74) * |
| Participants mothers' educational level | | | | |
| Unable to read and write | 311(32.7%) | 200(29.0%) | 1 | 1 |
| Able to read and write | 196(20.6%) | 217(31.4%) | 1.07 (.64 1.78) | 1.04(.62, 1.74) |
| Grade1-4 | 316(33.3%) | 212(30.7%) | 1.84 (1.10, 3.08) | 1.80(1.07, 3.02) |
| Grade5-8 | 5(1.0%) | 7(1.0%) | 1.11 (.67, 1.85) | 1.07(.64, 1.79) |
| Grade9-12 | 77(8.1%) | 27(3.9%) | 2.33 (.67, 8.08) | 2.13(.61, 7.46) * |
| College and above | 45(4.7%) | 27(3.9%) | .58 (.30, 1.11) | .58(.30, 1.12) |
| Information about sexual and reproductive health services | | | | |
| Yes | 713(75.1%) | 457(66.2%) | 1.53(1.23, 1.90) | 1.45(1.16, 1.80) * |
| No | 237(24.9%) | 233(33.8%) | 1 | |
| utilization of YFSRH services | | | | |
| Yes | 678(71.4%) | 348(50.4%) | 2.44(1.99,3.00) | 1.80(1.41, 2.30) |
| No | 272(28.6%) | 342(49.6%) | 1 | |
| Living arrangement of the respondent | | | | |
| With family | 157(16.5%) | 119(17.3%) | 1.12(.82, 1.52) | 1.13(.82, 1.54) |
| With partner | 145(15.3%) | 99(14.3%) | 1.01(.73, 1.39) | 1.13(.81, 1.58) |
| With friends | 188(19.8%) | 115(16.7%) | .90(.67, 1.22) | .90(.66, 1.23) |
| With employs/relative | 205(21.6%) | 185(26.8%) | 1.33(1.01,1.76) | 1.50(1.13, 2.00) * |
| Alone | 255(26.8%) | 172(24.9%) | 1 | 1 |
| Students ever had sexual intercourse | | | | |
| Yes | 713(75.1%) | 457(66.2%) | 1.53(1.23, 1.90) | 1.50(1.23, 1.96) * |
| No | 237(24.9%) | 233(33.8%) | 1 | |

*Significant at p-value<0.05; YFSRH = youth friendly sexual and reproductive health.

Debre Markos Town (36.9%) [23], Vietnam(36.7%) [7] and Ghana (72.8%) [26]. This might be due to demographic and cultural difference as well as difference in knowledge and attitude about sexual and reproductive health issues.

The result of multiple logistic regression models shown that participants' mothers who have attained secondary school were found to be 2 times more likely to have communication on sexual and reproductive health issues than these mothers who have unable to read and write. This study was consistent with study done in Harar, Mekele, Eastern Ethiopia and East Wollega zone [6,17,22,27]. This might be due to more educated women more likely to have better knowledge and access of social and print media exposure regarding to the importance of communication on sexual and reproductive health issues with their children.

In this study, the odds of having young-parent communication were higher among being secondary (grade 9&10) school students as compared to being primary school students. This finding was similar with studies conducted in Wolayita Zone [17]. This might be due to secondary school students are more likely to have better knowledge and not afraid to communicate about reproductive and sexual health issues with their parents. Students who have married were more likely to discuss about reproductive issues than students who have not married (single). This might be explained by married students might have better experiences on sexual and reproductive issues communication from their husbands.

Students who had information about sexual and reproductive health services were more likely to communicate to their parents than their counter parts. This finding was similar with the studies done Debre Markos Town [23]. This may be due to the fact that respondents who are more aware of SRH issues may be more eager to communicate and the information they have received may pave the way for communication to be initiated.

The study revealed that students who had lived with employs/relatives were more likely to have communicated to their parents than students who lived alone. This might be due to students gain advices and information about the risk of premarital sex from employs/relatives and this might be an input for students to communicate with their parent. Students who had sexual intercourse more likely to communicate with parent about sexual and reproductive issues than those never had sexual intercourse. This was in line with study conducted in Debre Markos [23]. This might be due to fear of complication that come after sexual intercourse.

## Strength

Community based study design and using large sample size can be taken as the strength of the study.

## Limitation of the study

The limitation would be due to sensitivity of the issues, social desirability bias and recall bias might be faced. Another limitation of this study focused on quantitative approach which could not address the "why" questions in detail.

## Conclusion and recommendation

In this study young-parent communications on sexual and reproductive health issues were low. Then, teachers and health extension workers should strengthen comprehensive SRH education for students in school, churches, mosques, health facilities and encouraging them to participate in different health clubs in school and outside school. Parent should give education for their children sexual and reproductive health during the era of young age. And also, to improve parent-adolescent communication, it is important to establish a national wide sexual and reproductive health information and enhancement of family planning for adolescents particularly for night students. Mixed quantitative and qualitative studies are better for further investigations to answer "why" questions.

## Supporting information

**S1 File. English version questionnaires.**
(DOCX)

## Acknowledgments

We are highly thanks to College of Health Sciences, Debre Tabor University, for giving written supportive letter for conducting this study. We would like to extend our thanks to Amhara Regional Education Bureau for giving supportive letter to conduct the study and providing the necessary preliminary information. We would also like to extend our appreciation to the study participants, supervisors and data collectors for their active participation. This study was conducted in accordance with the Declaration of Helsinki.

## Author Contributions

**Conceptualization:** Gedefaye Nibret Mihretie, Tewachew Muche Liyeh, Abeba Belay Ayalew.

**Data curation:** Gedefaye Nibret Mihretie, Habtamu Gebrehana Belay.

**Formal analysis:** Gedefaye Nibret Mihretie, Tewachew Muche Liyeh, Habtamu Gebrehana Belay, Habitamu Abe Tasew, Abeba Belay Ayalew.

**Funding acquisition:** Tewachew Muche Liyeh, Habitamu Abe Tasew.

**Investigation:** Gedefaye Nibret Mihretie, Habtamu Gebrehana Belay, Habitamu Abe Tasew.

**Methodology:** Gedefaye Nibret Mihretie, Habtamu Gebrehana Belay, Abeba Belay Ayalew.

**Project administration:** Gedefaye Nibret Mihretie, Tewachew Muche Liyeh.

**Resources:** Gedefaye Nibret Mihretie, Habtamu Gebrehana Belay.

**Software:** Gedefaye Nibret Mihretie, Yitayal Ayalew Goshu.

**Supervision:** Gedefaye Nibret Mihretie.

**Validation:** Gedefaye Nibret Mihretie, Tewachew Muche Liyeh.

**Visualization:** Gedefaye Nibret Mihretie, Yitayal Ayalew Goshu.

**Writing – original draft:** Gedefaye Nibret Mihretie, Yitayal Ayalew Goshu.

**Writing – review & editing:** Gedefaye Nibret Mihretie.

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
