## [Decision Letter · Decision Letter 0]

28 Jan 2021

PONE-D-20-39821

Young-Parent Communication on Sexual and Reproductive Health Issues among Young Female Night Students in Amhara Region, Ethiopia: Community-Based Cross-Sectional Study.

PLOS ONE

Dear Dr. Mihretie,

Thank you for submitting your manuscript to PLOS ONE. After careful consideration, we feel that it has merit but does not fully meet PLOS ONE’s publication criteria as it currently stands. Therefore, we invite you to submit a revised version of the manuscript that addresses the points raised during the review process.

We look forward to receiving your revised manuscript.

Kind regards,

Frank T. Spradley

Academic Editor

PLOS ONE

3. Please include additional information regarding the survey or questionnaire used in the study and ensure that you have provided sufficient details that others could replicate the analyses. For instance, if you developed a questionnaire as part of this study and it is not under a copyright more restrictive than CC-BY, please include a copy, in both the original language and English, as Supporting Information, or include a citation if it has been published previously.

4. In the Methods, please describe how the questionnaire was validated and/or pre-tested. If these did not occur, please provide the rationale for not doing so.

5. In statistical methods, please refer to any post-hoc corrections to correct for multiple comparisons during your statistical analyses. If these were not performed please justify the reasons. Please refer to our statistical reporting guidelines for assistance (https://journals.plos.org/plosone/s/submission-guidelines.#loc-statistical-reporting).

6. In your statistical analyses, please state whether you accounted for clustering by school or zone. For example, did you consider using multilevel models?

7. Thank you for stating the following financial disclosure:

"not applicable"

8. We note that you have indicated that data from this study are available upon request. PLOS only allows data to be available upon request if there are legal or ethical restrictions on sharing data publicly. For information on unacceptable data access restrictions, please see http://journals.plos.org/plosone/s/data-availability#loc-unacceptable-data-access-restrictions.

Reviewers' comments:

Reviewer's Responses to Questions

**Comments to the Author**

1. Is the manuscript technically sound, and do the data support the conclusions?

Reviewer #1: No

Reviewer #2: Yes

2. Has the statistical analysis been performed appropriately and rigorously? 

Reviewer #1: No

Reviewer #2: Yes

3. Have the authors made all data underlying the findings in their manuscript fully available?

Reviewer #1: Yes

Reviewer #2: Yes

4. Is the manuscript presented in an intelligible fashion and written in standard English?

Reviewer #1: No

Reviewer #2: Yes

5. Review Comments to the Author

Reviewer #1: Dear editor, I would like to say thank you for considering me to review the manuscript entitled as ‘Young-Parent Communication on Sexual and Reproductive Health Issues among Young Female Night Students in Amhara Region, Ethiopia

General comment

I have assessed your manuscript thoroughly and “I am sorry to say that it might not be considered for publication in the present form. I listed my comments below and I hope the comment would improve the manuscript and may allow a revised version to be reassessed otherwise not.

The scope of the study is not described clearly. Please clarify the detail and importance of Young-Parent Communication on Sexual and Reproductive Health Issues under introductions" Moreover the results of this study are somehow as expected, not new nor interesting, nor surprising: I do not believe this study adds anything interesting to the available knowledge from the literature. There are many papers describing the relationships between social status or cultural level or wealth or religious believes with adherence to Young-Parent Communication? What do the authors wish to add with their work? Is there any specific focus to be described within the specific area of interest in your study area? Please provide some extra reason for having specific interest in these findings. The abstract background particular is not focused and aim is not described enough. Please rephrase and detail more. Be focused please. The conclusion of the abstract is also not clear.

Major:

1. The Background is not well organized, and it missed some core parts that should have been addressed for a better justification of the current research. (1) First, the authors should tell us more about the importance of parent communication with their children in every aspects of their life. Does the government have ever conducted any policy interventions targeting to solve that reproductive issue? If yes what are the interventions? Secondly why is it interesting and important to focus on maternal healthcare service in Ethiopia? I wish to know the contextual motivations underlying your research.

Method, result, and discussion

1. What are the justifications for including those covariates in the model

2. Have the authors done: multicollinearity test, sensitivity test, and robustness test? If so, please add the results as an endnote

3. Using cross-sectional data would inevitably face the issue of endogeneity. Have the authors taken the endogeneity issue into consideration? For example, can you use the IV method to make your research findings more robust and convincing?

4. Results

Goodness of fit measures for estimated models should be provided in the regression table.

5. In the Discussion section, please give some detailed policy suggestions based on your findings.

Reviewer #2: This is a very interesting article that highlights the importance of communication between youth and their parents about sexual and reproductive health in African countries . However, the discussion section could be improved by reducing the repetition of absolute data in the results.

After these corrections, the article can be accepted.

6. PLOS authors have the option to publish the peer review history of their article (what does this mean?). If published, this will include your full peer review and any attached files.

Reviewer #1: No

Reviewer #2: No

---

## [Author Response · Author response to Decision Letter 0]

14 Apr 2021

Title: Young-Parent Communication on Sexual and Reproductive Health Issues among Young Female Night Students

 Dear editors, editor staffs and reviewers, thank you very much. 

We tried to correct and answer based on your comment and questions, respectively.

Reviewer 

Question 1: There are many papers describing the relationships between social status or cultural level or wealth or religious believes with adherence to Young-Parent Communication? What do the authors wish to add with their work? Is there any specific focus to be described within the specific area of interest in your study area?

Response: of course some research works have been done on day time students, there is no study conducted concerning young-parent communication on sexual and reproductive health issues focusing on young girls attending night school in Ethiopia generally and in the study area particularly. Night female students are high risk reproductive health problems like sexual violence, rape, STI including HIV infection and unintended pregnancy etc. 

Question 2: The abstract background particular is not focused and aim is not described enough. Please rephrase and detail more. 

Response: we tried to correct based on your comment

Question 3: The Background is not well organized, and it missed some core parts that should have been addressed for a better justification of the current research. 

Response: we tried to correct based on your comment

Question 4: The authors should tell us more about the importance of parent communication with their children in every aspects of their life. Does the government have ever conducted any policy interventions targeting to solve that reproductive issue? If yes what are the interventions? 

Response: Ethiopian government has identified the reproductive health of adolescents as one of the priority areas but not yet put in practice and there is little data available about night adolescent students.

Question 5: why is it interesting and important to focus on maternal healthcare service in Ethiopia?

Response: adolescent problems are not only short term but also lifelong negative impact later on maternal health (eg early motherhood, HIV infection, discounting from school etc). 

Question 6: What are the justifications for including those covariates in the model?

Response: to minimize confounding variables. 

Question 7: Have the authors done: multicollinearity test, sensitivity test, and robustness test? If so, please add the results as an endnote 

Response: We check that there is no multicollinearity effect between independent variables but we did not do sensitivity test and robustness test.

Question 8: Using cross-sectional data would inevitably face the issue of endogeneity. Have the authors taken the endogeneity issue into consideration? For example, can you use the IV method to make your research findings more robust and convincing?

 Response: the cross-section study design by itself is faced the issue of endogeneity, we did not consider instrumental variables.

Question 9: Goodness of fit measures for estimated models should be provided in the regression table.

Response: Hosmer-Lemeshow Goodness of model fit test was done. 

Question 10: In the Discussion section, please give some detailed policy suggestions based on your findings.

Response: we tried to correct based on your comment

---

## [Decision Letter · Decision Letter 1]

2 Jun 2021

Young-Parent Communication on Sexual and Reproductive Health Issues among Young Female Night Students in Amhara Region, Ethiopia: Community-Based Cross-Sectional Study.

PONE-D-20-39821R1

Dear Dr. Mihretie,

We’re pleased to inform you that your manuscript has been judged scientifically suitable for publication and will be formally accepted for publication once it meets all outstanding technical requirements.

Kind regards,

Frank T. Spradley

Academic Editor

PLOS ONE

Reviewers' comments:

Reviewer's Responses to Questions

**Comments to the Author**

1. If the authors have adequately addressed your comments raised in a previous round of review and you feel that this manuscript is now acceptable for publication, you may indicate that here to bypass the “Comments to the Author” section, enter your conflict of interest statement in the “Confidential to Editor” section, and submit your "Accept" recommendation.

Reviewer #2: All comments have been addressed

2. Is the manuscript technically sound, and do the data support the conclusions?

Reviewer #2: Yes

3. Has the statistical analysis been performed appropriately and rigorously? 

Reviewer #2: Yes

4. Have the authors made all data underlying the findings in their manuscript fully available?

Reviewer #2: No

5. Is the manuscript presented in an intelligible fashion and written in standard English?

Reviewer #2: Yes

6. Review Comments to the Author

Reviewer #2: I am satisfied with the answers given by the authors. I agree the publication of the paper with the amendments made.

7. PLOS authors have the option to publish the peer review history of their article (what does this mean?). If published, this will include your full peer review and any attached files.

Reviewer #2: No

---

## [Editor Report · Acceptance letter]

11 Jun 2021

PONE-D-20-39821R1 

Young-Parent Communication on Sexual and Reproductive Health Issues among Young Female Night Students in Amhara Region, Ethiopia: Community-Based Cross-Sectional Study. 

Dear Dr. Mihretie:

I'm pleased to inform you that your manuscript has been deemed suitable for publication in PLOS ONE. Congratulations! Your manuscript is now with our production department. 

Kind regards, 

on behalf of

Dr. Frank T. Spradley 

Academic Editor

PLOS ONE